# Investigation of Neurological Complications after COVID-19 Vaccination: Report of the Clinical Scenarios and Review of the Literature

**DOI:** 10.3390/vaccines11020425

**Published:** 2023-02-13

**Authors:** Wei-Ping Chen, Ming-Hua Chen, Shih-Ta Shang, Yung-Hsi Kao, Kuo-An Wu, Wen-Fang Chiang, Jenq-Shyong Chan, Hann-Yeh Shyu, Po-Jen Hsiao

**Affiliations:** 1Department of Internal Medicine, Taoyuan Armed Forces General Hospital, Taoyuan 325, Taiwan; 2Division of Neurology, Department of Internal Medicine, Taoyuan Armed Forces General Hospital, Taoyuan 325, Taiwan; 3Infectious Disease, Department of Internal Medicine, Taoyuan Armed Forces General Hospital, Taoyuan 325, Taiwan; 4Department of Life Sciences, National Central University, Taoyuan 320, Taiwan; 5Division of Pulmonary & Critical Care Medicine, Taoyuan Armed Forces General Hospital, Taoyuan 325, Taiwan; 6Division of Nephrology, Department of Internal Medicine, Taoyuan Armed Forces General Hospital, Taoyuan 325, Taiwan; 7Division of Nephrology, Department of Internal Medicine, Tri-Service General Hospital, National Defense Medical Center, Taipei 114, Taiwan

**Keywords:** COVID-19 vaccines, neurological complications, Bell’s palsy, Guillain—Barré syndrome, transverse myelitis, postvaccinal meningitis and encephalitis, vaccine-associated herpes simplex encephalitis, cerebral venous sinus thrombosis

## Abstract

Coronavirus disease 2019 (COVID-19), caused by severe acute respiratory syndrome coronavirus 2 (SARS-CoV-2), broke out in 2019 and became a pandemic in 2020. Since then, vaccines have been approved to prevent severe illness. However, vaccines are associated with the risk of neurological complications ranging from mild to severe. Severe complications such as vaccine-induced immune thrombotic thrombocytopenia (VITT) associated with acute ischaemic stroke have been reported as rare complications post-COVID-19 vaccination. During the pandemic era, VITT evaluation is needed in cases with a history of vaccination within the last month prior to the event. Cerebral venous sinus thrombosis (CVST) should be suspected in patients following immunization with persistent headaches who are unresponsive to analgesics. In this article, we investigated neurological complications after COVID-19 vaccination and provided more subsequent related clinical studies of accurate diagnosis, pathophysiological mechanisms, incidence, outcome, and management.

## 1. Introduction

Coronavirus disease 2019 (COVID-19) was declared a pandemic by the World Health Organization (WHO) in 2020. Vaccines mitigated COVID-19 outbreaks. According to the Centres for Disease Control and Prevention, COVID-19 vaccines are safe, and severe adverse effects are rare. Guillain—Barré syndrome (GBS) and thrombosis with thrombocytopenia syndrome (TTS) are rare complications of J&J/Janssen COVID-19 vaccination [1]. Currently, ChadOx1 nCoV-19 (Vaxzevria^®^; AstraZeneca; Cambridge, UK), BNT162b2 vaccine (Comirnaty^®^; Pfizer-BioNTech; New York, NY, USA; Mainz, Germany), mRNA-1273 vaccine (SPIKEVAX™; Moderna; Cambridge, MA, USA), and MVC-COV1901 (Medigen; Hsinchu; Taiwan) are four major vaccine types that are used in Taiwan. According to reports from the Taiwan Food and Drug Administration, there are several neurological adverse effects, such as cerebrovascular stroke, facial palsy, seizure, convulsion, febrile convulsion, transverse myelitis, acute disseminated encephalomyelitis, Guillain–Barre’ Syndrome, neuromyelitis optica, myelitis, encephalitis, aseptic meningitis, optic neuritis, and cerebral venous sinus thrombosis [2]. There have been limited case series pertaining to potentially rare neurological complications of COVID-19 vaccines in Taiwan, and we aimed to investigate the possible pathological mechanism of the adverse effects of these vaccines. In this article, we focused on cases of neurological complications after COVID-19 vaccinations in our hospital.

## 2. Materials and Methods

### 2.1. Study Design

This retrospective study included 19 patients who presented to Taoyuan Armed Forces General Hospital in Taiwan from 1 March 2021 to 31 August 2022. Data were retrieved by searching ICD 10 of T50. B95A (adverse effect of viral vaccine, initial encounter) and G51.0 (Bell’s palsy) in the electronic medical records. We included patients older than 18 years who received COVID-19 vaccines and presented with neurological complications. Patients were evaluated by neurologists. The associated clinical symptoms were recorded. Patients received neurological examinations, nerve conduction velocity examinations, and neuroimaging.

### 2.2. Diagnostic Workup for Neurological Complications

Facial motor impairment should be characterized as Bell’s palsy only after the exclusion of traumatic, iatrogenic, neoplastic, infectious, metabolic, and congenital aetiologies. Electroneuronography (ENoG) and electromyography (EMG) are the primary tests used for prognosticating facial nerve outcomes [3]. Peripheral neuropathy was diagnosed by symptoms and nerve conduction study (NCS) findings. Motor and sensory conduction velocity can be measured in peripheral nerves. In axonal neuropathies, conduction velocity is normal or reduced mildly and needle electromyography of denervation in affected muscles. In demyelinating neuropathies, nerve conduction velocities may be slowed considerably in affected fibres, or conduction is blocked completely [4]. The diagnosis of transverse myelitis was made based on “Proposed diagnostic criteria and nosology of acute transverse myelitis” [5]. The diagnosis was confirmed by T2 hyperintense signal change on spinal magnetic resonance imaging and exclusion of a compressive cord lesion. Lumbar puncture was administered for differential diagnosis. The diagnosis of encephalitis was made based on symptoms such as altered mental status, seizure, fever, focal neurological findings, and cerebrospinal fluid (CSF) analysis or neuroimaging [6]. Herpes encephalitis was diagnosed based on herpes simplex virus (HSV) polymerase chain reaction (PCR) analysis of CSF, which is the gold standard method for the diagnosis of HSV encephalitis [7]. CVST was confirmed by magnetic resonance venography (MRV) [8].

## 3. Results

Table 1 shows the sex, age, clinical presentations, onset time, type of vaccine, diagnosis, and patient outcomes. Table 2 shows the demographic characteristics of the included patients. The associated neurological manifestations were as follows: Bell’s palsy, peripheral neuropathy, transverse myelitis, seizure, encephalitis, CVST, and ICH. Cases were categorized as peripheral nervous system and central nervous system. Most adverse events were non-fatal.

### 3.1. Peripheral Nervous System

#### 3.1.1. Bell’s Palsy

Bell’s palsy was the most common neurologic complication among our patients. Six patients with Bell’s palsy received mRNA vaccines (five cases of mRNA1273 and one case of BNT 162b2), and one patient received the ChadOx1nCoV-19 vaccine. Three patients’ symptoms improved. Four patients were lost to follow-up. The symptoms occurred within 15 days after vaccination. The average age of our patients was 71 years old. The mean time from vaccination to the occurrence of Bell’s palsy was 5.4 (range, 2–15) days from the last dose. The index case is described here, and the other cases (cases 2–7) are presented in Table 1.

Case 1: A 68-year-old female presented with right facial weakness after the second dose of mRNA1273. Facial stimulation and blink tests indicated right facial neuropathy. She received thiamine, riboflavin, and prednisolone for treatment. Her symptoms resolved.

#### 3.1.2. Peripheral Neuropathy

Five patients had peripheral neuropathy. One patient received the ChadOx1nCoV-19 vaccine, and four patients had symptoms after receiving mRNA-1273.

Case 8: A 42-year-old female presented with painful tingling and burning sensations of the left sole to ankle region with radiation to the hip region and left foot numbness 5 days postChadOx1nCoV-19 administration. She received gabapentin, vitamin B, and prednisolone for treatment. The NCV and EMG study indicated left peroneal and tibial neuropathy. Her symptoms of pain resolved, but she had residual numbness in the left toes.

Case 9: A 37-year-old female had hypoesthesia over the left hand and foot and intermittent clonus over the bilateral feet and right face 6 days after mRNA-1273 vaccination. Brain MRI and MRA (MR angiography) were normal. The NCS was normal. Her symptoms resolved after clonazepam.

Case 10: A 58-year-old female had numbness and tingling pain over the right anterior tongue 6 days after the 3rd mRNA-1273 injection. Trigeminal neuralgia was impressed, and she received gabapentin for pain control. The patient’s condition improved partially after gabapentin.

Case 11: A 72-year-old male suffered from drop foot 13 days after the 2nd dose of mRNA-1273. The NCS indicated left deep peroneal neuropathy, neurapraxia, and right sural neuropathy. The patient was lost to follow-up.

Case 12: A 67-year-old male had a history of type 2 diabetes mellitus and hypertension. He had progressive numbness and shooting pain over the bilateral lower limbs. NCS showed motor polyneuropathy of the bilateral lower limbs, demyelination type. His symptoms resolved gradually.

### 3.2. Central Nervous System

#### 3.2.1. Transverse Myelitis (TM)

Acute transverse myelitis (ATM) was observed in two patients who received the third mRNA-1273 and the first dose of ChadOx1nCoV-19.

Case 13: A 60-year-old man had a history of stroke and hypertension. He had symptoms of numbness and soreness over the right lower leg below the right T10 dermatome 18 days after the 3rd mRNA-1273. He received spinal magnetic resonance imaging (MRI) and showed no significant cord signal change. Residual numbness persisted after treatment with steroids.

Case 14: A 56-year-old female had acute myelitis involving the left spinothalamic tract at the T7 level with right paresthesia below the T10 dermatome, and vaccine association was suspected. The symptoms occurred 8 days after the first dose of ChadOx1nCoV-19. Motor strength showed a Medical Research Council (MRC) grade of 5/5 in all four limbs. A lumbar puncture was administered. CSF analysis presented a normal glucose range (87.32 mg/dL), no pleocytosis (WBC < 5/µL), and normal protein levels (34.8 mg/dL). The IgG index was 0.76. The patient’s touch blood sugar level was 135 mg/dl (CSF/serum ratio: 0.64). The opening pressure was 20 cm H_2_O. CSF immune electrophoresis showed no obvious oligoclonal or monoclonal band. The serum aquaporin-4 antibody was negative. Multiple sclerosis and neuromyelitis optica were excluded. Laboratory data, including serum rheumatoid factor, serum complement C3 and C4, anti-Ro/La, anti-SM, anti-RNP, anti-cardiolipin IgM, anti-nuclear antibody, anti-neutrophil cytoplasmic antibody, thyroid function test, anti-HIV test, serum HSV 1 and HSV 2 IgM, EB VCA IgM, varicella zoster IgM, and rapid plasma reagin (RPR), were analyzed. The results were all in the normal range. MRI of the thoracolumbar spine revealed an ill-defined faint high T2 signal within the cord from T10 to the upper L1 level with minimal faint enhancement. Brain contrast-enhanced MRI revealed old lacunar infarction of the bilateral corona radiata, bilateral basal ganglia, and corpus callosum. Her symptoms gradually resolved.

#### 3.2.2. Postvaccinal Meningitis and Encephalitis

Three cases of postvaccinal encephalopathy were identified.

Case 15: A 22-year-old male [9] had generalized tonic—clonic convulsion 8 days after the second dose of mRNA-1273. He was intubated due to hypercapnic respiratory failure and persistent seizures. CSF analysis presented a normal glucose range (75.9 mg/dL), no pleocytosis (WBC < 5/µL), elevated protein levels (64.5 mg/dL), negative results of HSV-1, HSV-2 PCR, and venereal disease research laboratory (VDRL). The one-touch blood sugar level was 136 mg/dL (CSF/serum ratio: 0.55). The opening pressure was 38 cm H_2_O. No Mycobacterium tuberculosis complex, bacteria, or fungi were found on the culture. The IgG index was 0.59. The post-contrast brain MRI did not show evidence of leptomeningeal enhancement. Electroencephalography (EEG) indicated continuous moderate diffuse cerebral cortical dysfunction. He received methylprednisolone 1 g per day, which was gradually tapered off. After anti-epilepsy drugs and steroids, the patient still had several seizures after discharge.

Case 16: The second case was a 57-year-old male who had a history of schizophrenia. He had been taking medications such as zopiclone, quetiapine, chlorpromazine, and flunitrazepam inthe outpatient department. However, he did not take the medication regularly. He had symptoms of a first-time seizure 19 days after the first dose of mRNA-1273. CSF analysis presented a normal glucose range (90.1 mg/dL), no pleocytosis (WBC < 5/µL), elevated protein level (67 mg/dL), and negative HSV1,2 PCR and Cryptococcus Ag. His one-touch blood sugar was 135 mg/dL (CSF/serum ratio: 0.66). The opening pressure was 11 cm. The IgG index was 0.41. He received levetiracetam for seizure control. No seizures recurred after discharge.

Case 17: The third case was a 58-year-old female who had no previous systemic disease. She had symptoms of fever, shortness of breath, and disorientation 6 days after the third dose of the COVID-19 vaccine. She received two doses of ChadOx1nCoV-19, and the third dose was mRNA-1273. A lumbar puncture was performed. CSF analysis presented decreased glucose range (61.72 mg/dL), pleocytosis (WBC: 40/µL, lymphocyte predominant: 59%), elevated protein levels (82.9 mg/dL), and negative HSV1,2 PCR and Cryptococcus Ag. Her one-touch blood sugar was 153 mg/dL (CSF/serum ratio: 0.4). The opening pressure was 9 cm H_2_O. The IgG index was 0.32. Central nervous system (CNS) infection was ruled out based on CSF culture and virus analysis. Serum HSV-I, HSV-II IgG, IgM, CMV, and EB VCA IgM were negative. The RSV screen test was negative. A survey of autoimmune diseases and autoantibodies, including anti-thyroid peroxidase antibody (anti-TPO), anti-thyroglobulin antibody (ATA), anti-mitochondrial antibody, anti-nuclear antibody, and anti-ds DNA, was negative. Autoimmune disorder was ruled out. Her brain computed tomography (CT) revealed no abnormal findings. The post-contrast brain MRI did not show evidence of leptomeningeal enhancement. The patient received dexamethasone, which was gradually tapered off. Her symptoms resolved.

#### 3.2.3. Vaccine-Associated Herpes Simplex Encephalitis (HSE)

Case 18: A 64-year-old male had herpes simplex virus encephalitis. The patient had altered mental status and fever on the 6th day after the first dose of BNT 162b2. Blood laboratory investigation revealed a white blood cell count of 8240/µL (normal range, 4800–10,800), hemoglobin level of 12 g/dL (normal range, 12–16), and platelet count of 213,000/µL (normal range, 130,000–400,000) with a WBC differential including 73.2% neutrophils, 20.6% lymphocytes, 5.3% monocytes, and 0.5% eosinophils. Serum HSV IgM, CMV IgM, EB VCA IgM, varicella zoster IgM, rapid plasma reagin, Mycoplasma IgM, Chlamydia pneumoniae IgM, anti-TPO, ATA, urine pneumococcus rapid screen, and adenovirus Ag test via nasal swab were surveyed, and all presented negative findings. CSF analysis presented a mildly decreased glucose range (52 mg/dL), pleocytosis (WBC: 31/µL, lymphocyte predominant: 80%), elevated protein levels (54.3 mg/dL), and negative HSV2 PCR and Cryptococcus Ag. His blood sugar was 113 mg/dL (CSF/serum ratio 0.46). The opening pressure was 42 cm H_2_O. The IgG index was 0.79. India ink and Gram staining revealed no obvious pathogen. HSV 1 DNA was identified in the CSF PCR test (597 copies/mL). The EEG indicated continuous moderate diffused cerebral cortical dysfunction. The patient’s brain CT showed senile changes with cortical atrophy over bilateral frontotemporal regions with sulcal widening. Brain MRI revealed hyperintensity in the bilateral cingulate gyrus. The patient was diagnosed with HSE. The patient’s condition improved after treatment with acyclovir.

#### 3.2.4. Cerebral Venous Sinus Thrombosis (CVST) and Intracranial Hemorrhage (ICH)

The most severe neurological manifestations were cerebral venous sinus thrombosis and ICH.

Case 19: CVST was observed in one 40-year-old female [10]. She did not have any previous systemic disease. She presented with fever, headache, and petechiae 6 days after Chadox1 ncov-19 injection. The SARS-CoV-2 PCR test was negative. High D-dimer levels (>10,000 ng/mL; normal ≤ 250 ng/mL, latex-enhanced immunoturbidimetric immunoassay) and thrombocytopenia (31 × 109/L) were noted. The level of blood platelet factor 4 (PF4) antibodies was high (110.76 ng/mL; normal ≤ 40 ng/mL) and the result of the platelet activation test was positive, confirming the diagnosis of vaccine-induced immune thrombotic thrombocytopenia (VITT). No significant abnormal findings were found in the initial brain CT scan study. Brain MRV detected CVST. However, her headache worsened. Brain MRI with contrast was arranged. The right thalamus and left temporal region ICH were detected. This patient recovered gradually without neurological sequelae.

## 4. Discussion

Anti-severe acute respiratory syndrome coronavirus 2 (SARS-CoV-2) spike protein IgG detected in CSF and elevated serum spike-specific (Sp) SARS-CoV-2 IgG concentration may indicate that central nervous system adverse events were induced after vaccination [11]. SARS-CoV-2 spike protein, which is expressed after either mRNA or adenoviral-vector vaccines, binds to the angiotensin-converting enzyme 2 (ACE2) receptor followed by endocytosis. Neuropathologic findings may be due to direct endothelial cell damage caused by the SARS-CoV-2 spike protein which was due to molecular mimicry elicited by the spike protein [9]. The autoimmune response with complement activation and increased cytokine expression may cause disruption of blood–brain barrier and the concomitant thromboses/hypercoagulable state [12]. Thrombosis combined with thrombocytopenia after vaccination has been termed vaccine-induced prothrombotic immune thrombocytopenia (VIPIT), VITT, or thrombosis with thrombocytopenia syndrome (TTS). VITT has been mostly reported after immunization with adenovirus vector-based vaccines against COVID-19 [13,14,15,16,17]. The mean age of patients with VITT was 45.6 years (95% CI 43.8–47.4, I2 = 57%), with a female predominance (70%) [15]. Another systemic review presented a median age of 45.5 years [17]. One systemic review included 27 patients with the occurrence or relapse of thrombotic thrombocytopenic purpura (TTP) [16]. The mean age was 51.3 years. TTP was mostly observed after the BNT162b2 vaccine (after the first dose, n = 12; after the second dose, n = 7), followed by the mRNA-1273 vaccine (after the first dose, n = 2; after the second dose, n = 4). Adenoviral vaccines were responsible for four cases (Ad26.COV2-S-1, ChAdOx1 nCoV-19-3). These were the most common mechanisms of neurological complications of COVID-19 vaccines.

Age seems to be a poor prognostic factor of neurological adverse events. A systemic review noted that older age, higher modified Rankin scale (MRS), and second dose of vaccine presented a worse outcome in patients with COVID-19 vaccine myelitis [18]. According to the Sisonke’s study of Ad26.COV2.S, the median age of participants was 42 years and the majority were women. Adverse events decreased with increasing age (3.2% for age 18–30 years, 2.1% for age 31–45 years, 1.8% for age 46–55 years, and 1.5% for age > 55 years) [19]. The decline of innate immune defense mechanisms in older people may be the reason. Older patients tend to have lower systemic levels of IL-6, IL-10, C-reactive protein, and lower neutralizing antibody titers after vaccination as compared to younger individuals [20].

### 4.1. Peripheral Nervous System Disorder

#### 4.1.1. Bell’s Palsy

Inflammation and oedema of the facial nerve is the mechanism of facial nerve palsy. Inflammation-caused demyelination [21], anatomical variations of the fallopian canal, cold or viral prodrome, vasospasm-related primary ischaemia, transudate-associated secondary ischemia, thickening of the facial nerve sheath [22], HSV infection [23] and interferon [24] are possible etiologies of Bell’s palsy. The actual mechanism of Bell’s palsy after vaccination is unclear. Cytokine storm and direct invasion were the possible mechanisms that made SARS-CoV-2 the possible etiology of Bell’s palsy [25]. A sponsor (Pfizer) of the FDA’s Vaccines and Related Biologic Products Advisory Committee raised the possibility of innate immune activation from a combined effect of mRNA and lipids [26]. One case report disclosed a possible association between BNT162b2 and autoimmune reaction. IgG autoantibodies against type I IFNs increased compared toprevaccination, and second and third shots [27]. A higher frequency of facial paralysis after the Pfizer-BioNTech vaccine than after the Oxford-AstraZeneca vaccine was observed based on EudraVigilance data [28]. According to a large-scale meta-analysisthat disclosed a marginal increase in the incidence of Bell’s palsy after mRNA vaccinations compared to the unvaccinated groups, the incidence rate of Bell’s palsy was approximately 6.07 per 100,000 persons (95% CI, 3.49–8.65) after the first dose of mRNA vaccination and decreased to 5.06 per 100,000 persons (95% CI, 1.97–8.16) after the second dose of mRNA vaccination. The risk of Bell’s palsy after BNT162b2 was significantly higher than after mRNA-1273 [29]. Another cohort study reported that the incidence of Bell’s palsy was 15–101 per 100,000 person years in meta-analysis, dependingon age and sex strata [30]. In our case series report, most patients with Bell’s palsy received mRNA-1273.

#### 4.1.2. Guillain—Barré Syndrome (GBS)

There are some case reports of GBS being associated with the COVID-19 vaccines BNT162b2 [31] and ChadOx1nCoV-19 [32]. According to an interim analysis of surveillance of COVID-19 vaccines (Ad.26.COV2. S, BNT162b2, or mRNA-1273 COVID-19 vaccine), elevated risk of GBS after primary Ad.26.COV2. S was confirmed, and mRNA vaccines do not appear to be associated with GBS [33]. According to a meta-analysis, the incidence rate of GBS after mRNA vaccination was 0.19 per 100,000 persons (95% CI, 0.13–0.25) [29]. The risk of GBS of BNT162b2 was significantly higher than that of mRNA-1273 (POR, 2.85; 95% CI, 1.61–5.04) with low heterogeneity (I2, 0%). One multinational network cohort study estimated the incidence of GBS which ranged from one to twelve per hundred-thousand person-years (PY) in a meta-analysis stratified by age and sex [30]. The causal mechanism of GBS post-COVID-19 vaccines was not established. SARS-CoV-2 enters host cells via the S protein, which not only binds to the ACE2 receptor but also binds to sialic-acid-containing glycoproteins and gangliosides on cell surfaces [34]. Post-infectious immune-mediated mechanism was hypothesized because anti-ganglioside antibodies were recognized in several COVID-19 patients [35,36]. It may be hypothesized that SARS-CoV-2 spike-bearing gangliosides may form a neoantigen recognized by unidentified antibodies [36]. Cytokine storm was another possible mechanism of GBS induced by COVID [37]. Current treatments for GBS include immune globulin and plasma exchange [38]. Some patients died; others did not substantially improve [38]. In our case series, there were no Guillain—Barré syndrome cases associated with COVID-19 vaccines. This may be due to the limited sample size.

#### 4.1.3. Trigeminal Neuralgia

Trigeminal neuralgia is a rare neurological complication of COVID-19 vaccines. Immune-mediated inflammation is thought to be a more likely mechanism due to its rapid onset and response to corticosteroids and pregabalin [39,40,41]. Narasimhalu et al. reported a rare case of acute trigeminal neuritis and cervical radiculitis after Pfizer-BioNtech vaccination [40]. The 52-year-old female had a past medical history of diabetes mellitus, hypertension, hyperlipidemia, and scoliosis. Her symptoms started 3 h after the first dose of Pfizer-BioNtech vaccination and improved gradually after oral prednisolone. To date, the actual incidence of trigeminal neuralgia is not well known.

### 4.2. Central Nervous System

#### 4.2.1. Demyelinating Disease of the Central Nervous System

TM is inflammation of the spinal cord and causes acute or subacute neurologic signs of motor, sensory, or autonomic dysfunction. The demyelinating diseases of the central nervous system include multiple sclerosis (MS), myelin oligodendrocyte glycoprotein (MOG) antibody-associated disease (MOGAD), neuromyelitis optica spectrum disorder (NMOSD), and acute disseminated encephalomyelitis (ADEM) [42,43]. These disorders can cause TM [44]. NMOSD was reported after mRNA vaccines (86.56%), inactivated vaccines (11.94%), and viral vector vaccines (1.49%) [45]. MOGAD was less common. A 59-year-old male had MOGAD with longitudinal extensive transverse myelitis after ChadOx1nCoV-19 [46].

In case 14, the patient received high-dose steroids for 5 days, and her condition improved. This patient recovered partially. Case 13 responded poorly to steroids, which may be due to the late initiation of the steroid treatment. One multi-analysis included 37 cases of transverse myelitis following different vaccines including those against hepatitis B virus, measles–mumps, rubella vaccine, diphtheria–tetanus–pertussis (DTP), diphtheria–tetanus (DT), rabies vaccine, oral polio virus, influenza vaccine, typhoid vaccine, pertussis, and Japanese B encephalitis [47]. A multinational network cohort study estimated incidence rates of transverse myelitis. The incidence rangedfrom 1 to 4 per 100,000 PY in a meta-analysis, depending on age and sex strata [30]. Transverse myelitis associated with COVID-19 vaccines was observed recently. Pagenkopf and Sudmeyer reported a case of longitudinally extensive transverse myelitis following vaccination with ChadOx1nCoV-19. CSF analysis presented predominantly polymorphonuclear pleocytosis [48]. No pathogen was detected in CSF. There were no positive findings for anti-neuronal autoantibodies. A positive SARS-CoV-2-IgG serum antibody was detected without an IgG-antibody response against nucleocapsid antigen. Symptoms improved after high-dose corticoid therapy. The case report proved a close association between the COVID-19 vaccine and transverse myelitis. The inflammatory responseand autoimmunity due to cytokine storms have been proposed as possible causes of vaccine-related myelitis [47,49].

#### 4.2.2. Postvaccinal Meningitis and Encephalitis

A healthy 22-year-old man presented with a first seizure 6 days after the second dose of mRNA-1273 [9]. The patient received a lumbar puncture for CSF analysis. SARS-CoV-2 spike S1 receptor-binding domain (RBD) IgG was detected in CSF at a concentration of 98.54 binding antibody units (BAU)/mL and in serum at a concentration of 3270.22 BAU/mL. SARS-CoV-2 spike S1 domain antibody in CSF suggests a possible association with postvaccine complications [9]. There have been some case reports of autoimmune encephalitis involving ChadOx1nCoV-19 [50,51] and BNT162b2 [52,53]. Zuhorn et al. reported a case series of three patients whose onset of encephalitis was within seven to eleven days [51]. The temporal association is in line with our case. Baldelli et al. reported a case of hyperacute encephalopathy following ChAdOx1 nCoV-19 vaccine administration [50]. Cytokine levels documented a significant increase in interleukin (IL)-6 in both CSF and serum and increased IL-8 in CSF. Saito et al. reported a case of a 42-year-old female who had severe headache and fever after vaccination with BNT162b2. Sp SARS-CoV2 IgG was detected (17.1 AU/mL) in CSF [53]. Elevated serum Sp SARS-CoV-2 IgG (393.4 AU/mL; cut-off 0–50 AU/mL; (Chemiluminescent Enzyme Immunoassay; Architect Quant IgG II, Abbott) was noted. These cases responded well to systemic steroids. Vojdania and Kharrazian conducted research that measured the degree of immune reactivity of anti-SARS-CoV-2 spike protein monoclonal antibody and anti-SARS-CoV-2 nucleoprotein monoclonal antibody with human antigens [54]. Vojdania et al. further applied human monoclonal anti-SARS-CoV-2 antibodies (spike protein, nucleoprotein) and rabbit polyclonal anti-SARS-CoV-2 antibodies (envelope protein, membrane protein) to 55 different tissue antigens [55]. Human anti-SARS-CoV-2 spike protein antibody reacted strongest with neurofilament protein (NFP), followed by mitochondrial M2 antigen, glutamic acid decarboxylase 65 (GAD-65), and nuclear antigen [55]. This study found immune reactivity between SARS-CoV-2 antibodies and barrier target proteins (occludin + zonulin, beta-catenin, and S100B), which were responsible for the integrity of the barriers. Hussain et al. generated an adult mouse model and proved that suppressing endothelial Wnt/β-catenin signaling causes increased blood—brain barrier permeability [56]. This demonstrated that β-catenin-dependent Wnt signaling maintains the integrity of the adult blood—brain barrier. CTNNB1 encodes β-catenin. Tran et al. used an inducible and conditional knockout (icKO) mouse model with tamoxifen-inducible endothelial cell-restricted disruption of CTNNB1, and the iCKO mice developed severe seizures, neuronal injury, and central nervous system inflammation [57]. The immune reactivity between SARS-CoV-2 antibodies and brain-barrier protein is responsible for the neurological complications of vaccination. There are some mechanisms of COVID-19 encephalitis. Autoimmune encephalitis and systemic inflammatory status responses aretwo of the same mechanisms between vaccine and SARS-CoV-2 related encephalitis [58,59]. Direct virus invasion was the different mechanism of SARS-CoV-2-induced encephalitis from postvaccinalencephalitis. SARS-CoV-2 PCR positivity has been detected in the CSF in a small number of patients [60]. The resulting incidence of encephalitis following ChadOx1nCoV-19 was 0.08 per 100,000 PY and 0.02 following BNT162b2 [51]. It is essential to recognize and provide appropriate treatment for this complication.

#### 4.2.3. Vaccine-Associated Herpes Simplex Encephalitis

Several case reports have revealed the effects of SARS-CoV-2 vaccinations on herpes simplex virus and varicella-zoster virus reactivation [61,62,63,64,65,66,67,68,69]. Cases of herpetic meningoencephalitis following COVID-19 vaccines of ChAdOx1 nCoV-19 [70] and Covaxin [71] were reported. However, the actual incidence of vaccine-associated herpes simplex encephalitis was not well established. Transient lymphopenia was observed in participants vaccinated with BNT162b1 [72]. Toll-like receptors, plasmacytoid dendritic cells, interferon, natural killer cells, macrophages [73], and CD4 and CD8 T cells [74] contribute to the immune response against HSV [75]. Lymphocyte depletion may be related to primary herpes or reactivation of latent herpes [76,77]. The immunomodulation process has been reported as a possible cause of herpes virus reactivation after vaccination [78]. A higher frequency of herpes reactivation was observed during COVID-19. This may be explained by immune dysregulation including CD4 cell and NK cell cytopenia, and stimulation of IL-6 and tumor necrosis factor-α [79]. Further studies on the mechanism of susceptibility to herpesvirus infection after COVID-19 vaccination can help clarify the association.

#### 4.2.4. Stroke

The COVID-19 vaccine causes platelet-activating antibodies against platelet factor 4 (PF4). The onset of thrombosis at unusual sites such as in the brain or abdomen and thrombocytopenia occurring 5 to 20 days after vaccination can present a rare adverse effect of ChAdOx1 nCov-19 vaccination [80]. The detailed molecular pathway was described by Goldman M, and Hermans C [13]. Positive-charge PF4 bound to the negative surface of adenoviral vector hexons forms an immunogenic complex and triggersanti-PF4 antibodies. Macrophages, monocytes, natural killer cells, and dendritic cells take up antibody-PF4 (adenovirus) complexes via FC gamma receptors, resulting in the activation of the cells. Platelets bind to this complex via their Fc gamma RIIA and activate the release of PF4 which results in thrombosis. Another mechanism was related to soluble spike protein. Soluble spike protein binds to the ACE2 receptor and may trigger immunological events and induce thromboembolic events [81]. This mechanism has been termed “Vaccine-Induced COVID-19 Mimicry” syndrome and may explain the correlation between the adenovirus-based vaccine and thromboembolic events [81]. SARS-CoV-2 and adenovirus bind and activate various toll-like receptors (TLRs), leading to activation of the NFκB signaling pathway and further releasing phosphorylated p50/p65 heterocomplex to the nucleus where transcription of genes including TNFα, IL-1, IL-6, MCP-1, MCP-3, ICAM-1, VCAM-1, complement components and coagulation factors, such as PAI-1 is induced [82,83,84]. ACE2 activity is lost during SARS-CoV-2 viral entry [85]. The loss of ACE2 activity allows Ang II-mediated activation of p38 MAPK activation and simultaneously promotes thrombosis [86]. The P38 MAPK pathway regulates the translation of TNF-α and IL-1β which also activates the NF-κB pathway [87]. The NF-κB pathway is involved in inflammatory and thrombotic responses [88]. Ischaemicstroke, haemorrhagicstroke, and CVST were reported after COVID-19 vaccines [89]. Ischaemicstrokes are associated with VITT. Haemorrhagicstroke isusually secondary to CVST [90]. One case report revealed a case without cardiovascular risk factors, and with no remarkable past medical history that suffered from a large primary hemorrhagic stroke after ChAdOx1 nCoV-19 vaccination without thrombocytopenia, thrombosis, or aneurysm [91]. VITT was not the only mechanism of vaccine-related ICH. CVST is a rare form of stroke and often occurs in young and middle-aged women [92]. According to an analysis of cases reported to the European Medicines Agency about CVST [93], more cases were reported after ChAdOx1 nCov-19 vaccination than after mRNA vaccination. Antibodies against PF4 were reported in 8% of cases in the ChAdOx1 nCov-19 group but in none in the mRNA vaccination group. Thrombocytopenia was reported in 57% of CVST cases in the ChAdOx1 nCov-19 group. CVST occurring after ChAdOx1 nCov-19 has different associated laboratory data from the mRNA vaccination group [93]. Thrombocytopenia was only reported in the ChAdOx1 nCov-19 group and pre-COVID-19 CVST group. No case was reported in the mRNA vaccine group. Kowarz et al. reported molecular evidence that vector-based vaccines were associated with soluble spike protein variants [81]. Production of soluble spike protein was caused by splicing events. Soluble spike protein has been found to cause an inflammatory response in endothelial cells [12,94]. A population-based cohort study estimated the risk of CVST after Ad26.COV2.S [94]. The overall age- and sex-adjusted CVST incidence post-Ad26.COV2.S vaccination was 2.34 per 100,000 PY. Age-adjusted CVST rates for female and male individuals were 2.46 per 100,000 PY and 2.34 per 100,000 PY, respectively. Another web-based questionnaire focused on the incidence of cerebral sinus and venous thrombosis within one month after COVID-19 vaccination in 2021 in nine German states [95]. The incidence rate was 1.52 (95% CI = 1.00–2.21) per 100,000 person-months for ChAdOx1, 0.11 per 100,000 person-months for BNT162b2, and 0.00 (95% CI = 0.00–1.48) per 100,000 person-months for mRNA-1273. Although CVST is rare, prompt identification and initiated treatment are crucial. CVST with VITT appeared to have higher rates of comaand ICH, and a high mortality rates [96]. Current treatment for VITT includes high-dose immune globulin, corticosteroids, rituximab, eculizumab, and therapeutic plasma exchange. Non-heparin anticoagulants, including fondaparinux, direct oral anticoagulants (DOACs), and direct thrombin inhibitors (argatroban, bivalirudin) were suggested [97].

Our study has some limitations. First, the small sample size does not represent thepopulation of vaccine complications. This observational study lacked baseline incidence due to the lack of the total number of individuals vaccinated near our region. Most incidences of Bell’s palsy presented after the first and second doses [29]. It is not common for adverse events to appear after the fourth dose of the vaccine. Only one of our cases had Bell’s palsy after the fourth dose of vaccination. This finding is consistent with other reports.

## 5. Conclusions

Although most neurological complications are non-fatal, there are some fatal complications, such as CVST, GBS, ADEM, and ICH. Both COVID-19 and spike protein-related vaccines produce immunological complications through the same mechanism of molecular mimicry. VITT is another unique mechanism. Clinicians should be aware of these possible complications and perform a throughdifferential diagnoses. Early recognition and appropriate treatments should be initiated once these adverse events occur. Future investigations will be required to identify the risk factors for neurological complications of different COVID vaccines and technologically advanced new vaccines against unfavourableadverse effects.

## Figures and Tables

**Table 1 vaccines-11-00425-t001:** Patients with neurological complications after receiving COVID-19 vaccines.

Case No.	Age Sex	Symptoms	Vaccine	Underlying Disease	Onset Time	Diagnosis	Outcome
1	68 F	Right facial weakness and tingling pain, numbness over left face	1st, 2nd: mRNA-1273	Coronary artery disease	5 days	Bell’s palsy	Resolution of facial weakness and tingling pain
2	63 M	Left facial weakness, pain, drooling	1st: ChadOx1nCoV-19	Diabetes mellitus Hypertensive cardiovascular disease Dyslipidemia	10 days	Bell’s palsy	Failed to follow up
3	80 F	Left facial weakness	1st: BNT 162b2	Type 2 diabetes mellitus Hypertension	14 days	Bell’s palsy	Failed to follow up
4	57 M	Right facial weakness	1st, 2nd, 3rd: mRNA-1273	None	2 days	Bell’s palsy	Failed to follow up
5	71 F	Left face weakness	4th mRNA-1273	None	2 days	Bell’s palsy	Partial improvement. Mild weakness of left lower face
6	79 F	Right face weakness	1st, 2nd: mRNA-1273	Type 2 diabetes mellitus	15 days	Bell’s palsy	Resolution of right face weakness
7	80 M	Weakness of right face 3 days after 2nd mRNA-1273	1st, 2nd: mRNA-1273	Type 2 diabetes mellitus Hypertension Abducens palsy of right due to diabetic neuropathy Prostate cancer	3 days	Bell’s palsy	Failed to follow up
8	42 F	Electrical and burning painful sensation of left sole to ankle region with radiation to hip region Left foot numbness	1st: ChadOx1nCoV-19	None	5 days	Left peroneal and tibial neuropathy	Resolution of left sole pain Residual numbness of left toes
9	37 F	Hypoesthesia over left hand and foot, intermittent clonus over bilateral feet, and right face for days after mRNA-1273 vaccination	1st: ChadOx1nCoV-19 2nd: mRNA-1273	Ankylosing spondylitis	6 days	Myoclonus	Partial improvement
10	58 F	Numbness and tingling pain over right anterior tongue	3rd mRNA-1273	Hypertension	6 days	Trigeminal neuralgia	Partial improvement
11	72 M	Left foot dropped after 2nd mRNA-1273	1st, 2nd: mRNA-1273	Hypertension	13 days	Left deep peroneal neuropathy	Failed to follow up
12	67 M	Progressive numbness and shooting pain over bilateral lower limbs	1st: mRNA-1273	Type 2 diabetes mellitus Hypertension	3 days	Motor polyneuropathy of bilateral lower limbs, demyelination type	Resolution of numbness
13	60 M	Numbness, soreness over right lower leg below right T10 dermatome	3rd mRNA-1273	Stroke Hypertension	18 days	Possible acute myelitis	Residual numbness of right lower leg
14	56 F	Numbness with burning sensation in right lower limb below inguinal region	1st: ChadOx1nCoV-19	Type 2 diabetes mellitus Hypertension Hyperlipidemia	5 days	Acute myelitis involving left spinothalamic tract at T7 level with right paraesthesia below T10 dermatome, suspected vaccine-associated	Partial improvement Residual burning sensation in right lower limb
15	22 M	Fever, seizure	2nd: mRNA-1273	None	8 days	Vaccine-induced encephalitis and status epilepticus	Partial improvement. Recurrent seizures
16	57 M	First acute seizure	1st: mRNA-1273	Schizophrenia	19 days	Aseptic meningitis with seizure	Complete remission
17	58 F	Fever, altered mental status	1st, 2nd: ChadOx1nCoV-19 3rd: mRNA-1273	None	6 days	Aseptic encephalitis secondary to mRNA-1273	Complete remission
18	64 M	Fever, altered mental status, nausea, and vomiting	1st: BNT 162b2	Major depressive disorder	9 days	Vaccine-associated Herpes simplex encephalitis	Complete remission
19	43 F	Fever, headache, petechiae, mild shortness of breath	ChadOx1nCoV-19	None	9 days (cerebral venous sinus thrombosis, pulmonary embolism) 26 days (Intracranial hemorrhage) 32 days (Right hepatic thrombosis)	1. Vaccine induced thrombosis and thrombocytopenia 2. Cerebral venous thrombosis 3. Acute cerebral hemorrhage, right thalamus and left temporal lobe	Complete remission

**Table 2 vaccines-11-00425-t002:** Demographic characteristics of patients who had neurological complications following COVID-19 vaccines in AFTYGH.

Demographic Data	Bell’s Palsy			Peripheral Neuropathy		ATM		Post Vaccinal Meningitis and Encephalitis		Vaccine-Associated HSE	CVST
Vaccine types	AZ N = 1	Moderna N = 5	BNT N = 1	AZ N = 1	Moderna N = 4	AZ N = 1	Moderna N = 1	Moderna N = 2	AZ and Moderna N = 1	BNT N = 1	AZ N= 1
Male	1	2	0	0	2	0	1	2	0	1	0
Female	0	3	1	1	2	1	0	0	1	0	1
Mean onset after vaccination	10	5.4(2–15)	14	5	7 (3–13)	5	18	13.5(8–19)	6	9	9
Mean age	63	71	80	42	58.5 (37–72)	56	60	39.5(22–57)	58	64	43
Age categories											
<50	0	0	0	1	1	0	0	1	0	0	1
≥50	1	5	1	0	3	1	1	1	1	1	0

Abbreviation: AZ: ChadOx1nCoV-19; Moderna: mRNA-1273; BNT: BNT 162b2; HSE: herpes simplex encephalitis; ATM: acute transverse myelitis; CVST: cerebral venous sinus thrombosis.

## Data Availability

The data presented in this study are available on request from the corresponding author. The data are not publicly available due to privacy restrictions.

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
