# Peer review of "Investigation of Neurological Complications after COVID-19 Vaccination: Report of the Clinical Scenarios and Review of the Literature"

_vaccines, 2023, doi:10.3390/vaccines11020425_

Round 1
Reviewer 1 Report
The title of the manuscript is misleading. It is not impossible to describe the clinical manifestation of COVID-19 vaccination complications with 19 case reports. The strength of the manuscript is the Discussion. Call it "Case reports of c-19 vaccination complications and review of the literature". Since there have been millions of vaccinations and just a few neurological complications, statements of frequency and the rarity of those are needed.
The manuscript describes some cases with neurological complications, likely to be temporarily associated with COVID-19 vaccinations. In Discussion, neurological complications of vaccinations in the central and peripheral nervous system are described, with a couple of thoughts as to pathophysiology.
Strengths: Confirmation that COVID-19 vaccinations rarely have neurological complications in the central and peripheral nervous systems.
Weaknesses:
a) In which cohort/population have the complications been observed?
b) How was association to the vaccination made likely: temporal neighbourship only? Antibody explorations? Cerebrospinal fluid examinations including antibodies? Other laboratory tests including spike protein?
c) How complete is the listing of complications in the Discussion? Did you search the complete world literature?
d) Conclusion is that neurological complications are rare in general. However, there are no frequencies mentioned either in your scarcely described cohort or in general literature. Since SARS-related diseases are endothelial diseases, with an increase in stroke and myocardial infarction possibly related to activation of ACE receptors, it is also necessary to discuss pathophysiological aspects: how likely is it, that both COVID-19 and spike protein-related vaccination complications produce vascular and immunological complications via the same mechanism?
Author Response
Response to Reviewer 1 Comments
The title of the manuscript is misleading. It is not impossible to describe the clinical manifestation of COVID-19 vaccination complications with 19 case reports. The strength of the manuscript is the Discussion. Call it "Case reports of c-19 vaccination complications and review of the literature". Since there have been millions of vaccinations and just a few neurological complications, statements of frequency and the rarity of those are needed.
Author Reply
We are deeply honored by the time and effort you spent in reviewing this manuscript. We have revised the manuscript thoroughly according to your suggestions. The title was also changed to “Investigation of Neurological Complications after COVID-19 Vaccination: Report of the Clinical Scenarios and Review of the Literature”. We added more literature review sections for the rarity and frequency of these neurological complications (the detail information is in reply for point 1). All changes have been marked on yellow background. The responses to your comments are below.
Point 1: In which cohort/population have the complications been observed?
Author Reply
Thank you for the suggestion. We have amended the manuscript as follows:
In the discussion paragraph, we added the related cohort studies which observed the complications.
The incidence rate of Bell’s palsy was about 6.07 per 100,000 persons (95% CI, 3.49–8.65) after the first dose of mRNA vaccination and decreased to 5.06 per 100,000 persons (95% CI, 1.97–8.16) after the second dose of mRNA vaccination.
The incidence rate of Guillain‒Barré Syndrome (GBS) after mRNA vaccination was 0.19 per 100,000 persons (95% CI, 0.13–0.25) [31]. Narasimhalu et al. reported a rare case of acute trigeminal neuritis and cervical radiculitis after Pfizer-BioNtech vaccination [42].
The actual incidence of trigeminal neuralgia was not well known. Demyelinating disease of the central nervous system include multiple sclerosis (MS), myelin oligodendrocyte glycoprotein (MOG) antibody associated disease (MOGAD), neuromyelitis optica spectrum disorder (NMOSD), and acute disseminated encephalomyelitis (ADEM) [44, 45]. These disorders can cause TM [46]. NMOSD was reported after mRNA vaccines (86.56%), inactivated vaccines (11.94%), and viral vector vaccines (1.49%) [47]. MOGAD was less commonly. A 59-year-old male had MOGAD with longitudinal extensive transverse myelitis after ChadOx1nCoV-19 [48]. A multinational network cohort study estimated incidence rates of transverse myelitis. The incidence was ranging from 1-4 per 100,000 PY in meta-analysis, depend on age and sex strata [32]. Transverse myelitis associated with COVID-19 vaccines was observed recently. Pagenkopf and Sudmeyer reported a case of longitudinally extensive transverse myelitis following vaccination with ChadOx1nCoV-19. CSF analysis presented predominantly polymor-phonuclear pleocytosis [50]. No pathogen was detected in CSF. There were no positive findings for anti-neuronal autoantibodies. A positive SARS-CoV-2-IgG serum antibody was detected without an IgG-antibody response against nucleocapsid antigen. Symptoms improved after high dose corticoid therapy. The case report proved a close association between the COVID-19 vaccine and transverse myelitis. Inflammatory response and autoimmunity due to cytokine storm have been proposed as a possible cause of vaccine related myelitis [49, 51]. A population-based cohort study estimated the risk of CVST after Ad26.COV2.S [98]. The overall age- and sex-adjusted CVST incidence post–Ad26.COV2.S vaccination was 2.34 per 100,000 PY. Age-adjusted CVST rates for female and male individuals were 2.46 per 100,000 PY and 2.34 per 100,000 PY, respectively. Another web‐based questionnaire focused on the incidence of cerebral sinus and venous thrombosis within one month after COVID‐19 vaccination in 2021 in 9 German states [99]. The incidence rate was 1.52 (95% CI = 1.00–2.21) per 100,000 person‐months for ChAdOx1, 0.11 per 100,000 person‐months for BNT162b2, and 0.00 (95% CI = 0.00–1.48) per 100,000 person‐months for mRNA‐1273.
Point 2: How was association to the vaccination made likely: temporal neighborship only? Antibody explorations? Cerebrospinal fluid examinations including antibodies? Other laboratory tests including spike protein?
Author Reply
Thank you for the suggestions and comments for pathophysiology of neurological complications of COVID-19 vaccination. We agree with your comment and have amended the manuscript as below:
Anti- severe acute respiratory syndrome coronavirus 2 (SARS-CoV-2) spike protein IgG detected in CSF and elevated serum Spike-specific (Sp) SARS-CoV-2 IgG concentration may indicated that the central nervous system adverse events were induced after vaccination [11, 12]. SARS-CoV-2 spike protein which expressed after either mRNA or adenoviral-vector vaccines, binds to the angiotensin-converting enzyme 2 (ACE2) receptor followed by endocytosis. Neuropathologic findings may be due to direct endothelial cell damage cause by SARS-COV-2 spike protein which was due to molecular mimicry elicited by the spike protein [13]. The autoimmune response with complement activation and increased cytokine expression may cause disruption of blood-brain barrier and the concomitant thromboses/hypercoagulable state [14]. Thrombosis combined with thrombocytopenia after vaccination has been termed vaccine-induced prothrombotic immune thrombocytopenia (VIPIT), VITT or thrombosis with thrombocytopenia syndrome (TTS). VITT has been reported after immunization with adenovirus vector-based vaccines against COVID-19 mostly [15-19]. The mean age of patients with VITT was 45.6 years (95% CI 43.8-47.4, I2=57%), with a female predominance (70%) [17]. Another systemic review presented with the median age of 45.5 years [19]. One systemic review which included 27 patients with occurrence or relapse of Thrombotic thrombocytopenic purpura (TTP) [18]. The mean age was 51.3 years. TTP was mostly seen after BNT162b2 vaccine (after first dose, n = 12; after second dose, n = 7), followed by mRNA-1273 vaccine (after first dose, n = 2; after second dose, n = 4). Adenoviral vaccines were responsible for 4 cases (Ad26.COV2-S - 1, ChAdOx1 nCoV-19 - 3). These were the most common mechanisms of neurological complications of COVID-19 vaccines.
Point 3: How complete is the listing of complications in the Discussion? Did you search the complete world literature?
Author Reply
Thank you for the suggestion. We have amended the comprehensive literature as follows:
4.1. Peripheral Nervous System Disorder
4.1.1 Bell’s Palsy
Inflammation and oedema of the facial nerve is the mechanism of facial nerve palsy. Inflammation-caused demyelination [23], anatomical variations of fallopian canal, cold or viral prodrome, vasospasm related primary ischaemia, transudate associated secondary ischaemia, thickening of facial nerve sheath [24], HSV infection [25] and Interferon [26] are possible aetiologies of Bell’s palsy. The actual mechanism of Bell’s palsy after vaccination is unclear. Cytokine storm and direct invasion were the possible mechanism which made SARS-COV-2 the possible aetiology of Bell’s palsy [27]. There were some FDA's Vaccines and Related Biologic Products Advisory Committee, and a sponsor (Pfizer) raised the possibility of innate immune activation from a combined effect of mRNA and lipids [28]. One case report disclosed a possible association between BNT162b2 and autoimmune reaction. IgG autoantibodies against type I IFNs increased compared to pre-vaccination, second and third shots [29]. A higher frequency of facial paralysis after the Pfizer-BioNTech vaccine than after the Oxford-AstraZeneca vaccine was observed based on EudraVigilance data [30]. According to a large-scale meta-analysis which disclosed a marginal increase of incidence of Bell’s palsy after mRNA vaccinations compared to the unvaccinated groups, the incidence rate of Bell’s palsy was about 6.07 per 100,000 persons (95% CI, 3.49–8.65) after the first dose of mRNA vaccination and decreased to 5.06 per 100,000 persons (95% CI, 1.97–8.16) after the second dose of mRNA vaccination. The risk of Bell’s palsy after BNT162b2 was significantly higher than after mRNA-1273 [31]. Another cohort study reported that the incidence of Bell’s palsy was 15-101 per 100,000 person years in meta-analysis, depend on age and sex strata [32]. In our case series report, most patients with Bell’s palsy received mRNA-1273.
4.1.2 Guillain‒Barré Syndrome (GBS)
There are some case reports of GBS being associated with the COVID-19 vaccines BNT162b2 [33] and ChadOx1nCoV-19 [34]. According to an interim analysis of surveillance of COVID-19 vaccines (Ad.26.COV2. S, BNT162b2, or mRNA-1273 COVID-19 vaccine), elevated risk of GBS after primary Ad.26.COV2. S was confirmed, and mRNA vaccines do not appear to be associated with GBS [35]. According to a meta-analysis, the incidence rate of GBS after mRNA vaccination was 0.19 per 100,000 persons (95% CI, 0.13–0.25) [31]. The risk of GBS of BNT162b2 was significantly higher than that of mRNA-1273 (POR, 2.85; 95% CI, 1.61–5.04) with low heterogeneity (I2, 0%). One multinational network cohort study estimated the incidence of GBS which ranged from 1-12 per 100,000 per-son-years (PY) in meta-analysis stratified by age and sex [32]. The casual mechanism of GBS post COVID-19 vaccines was not established. SARS-COV-2 entered host cells via S protein, which not only binds to ACE2 receptor but also binds to sialic acid–containing glycoproteins and gangliosides on cell surfaces [36]. Post-infectious immune-mediated mechanism was hypothesized because anti-ganglioside antibodies were recognized in several COVID-19 patients [37, 38]. It may be hypothesized that the SARS-CoV-2 spike-bearing gangliosides may form a neoantigen recognized by unidentified antibodies [38]. Cytokine storm was another possible mechanism of GBS induced by COVID [39]. Current treatment for GBS include immune globulin and plasma exchange [40]. Some patients died; another didn’t substantially improve [40]. In our case series, there were no Guillain‒Barré syndrome cases associated with COVID-19 vaccines. This may be due to limited by sample size.
4.1.3 Trigeminal Neuralgia
Trigeminal neuralgia is a rare neurological complication of COVID-19 vaccines. Immune-mediated inflammation is thought to be a more likely mechanism due to its rapid onset and response to corticosteroids and pregabalin [41-43]. Narasimhalu et al. reported a rare case of acute trigeminal neuritis and cervical radiculitis after Pfizer-BioNtech vaccination [42]. The 52-year-old female had a past medical history of diabetes mellitus, hypertension, hyperlipidaemia, and scoliosis. Her symptoms started 3 hours after the first dose of Pfizer-BioNtech vaccination and improved gradually after oral prednisolone. To date, the actual incidence of trigeminal neuralgia is not well known.
4.2. Central Nervous System
4.2.1 Demyelinating disease of the central nervous system
Transverse myelitis (TM) is inflammation of the spinal cord and causes acute or subacute neurologic signs of motor, sensory or autonomic dysfunction. Demyelinating disease of the central nervous system include multiple sclerosis (MS), myelin oligodendrocyte glycoprotein (MOG) antibody associated disease (MOGAD), neuromyelitis optica spectrum disorder (NMOSD), and acute disseminated encephalomyelitis (ADEM) [44, 45]. These disorders can cause TM [46]. NMOSD was reported after mRNA vaccines (86.56%), inactivated vaccines (11.94%), and viral vector vaccines (1.49%) [47]. MOGAD was less commonly. A 59-year-old male had MOGAD with longitudinal extensive transverse myelitis after ChadOx1nCoV-19 [48]. In case14, the patient received high-dose steroids for 5 days, and her condition improved. This patient recovered partially. Case 13 responded poorly to steroids, which may be due to late initiation of the steroid treatment. One multi-analysis included 37 cases of transverse myelitis following different vaccines including those against hepatitis B virus, measles-mumps, rubella vaccine, diphtheria–tetanus–pertussis (DTP), diphtheria–tetanus (DT), rabies vaccine, oral polio virus, influenza vaccine, typhoid vaccine, pertussis, and Japanese B Encephalitis [49]. A multinational network cohort study estimated incidence rates of transverse myelitis. The incidence was ranging from 1-4 per 100,000 PY in me-ta-analysis, depend on age and sex strata [32]. Transverse myelitis associated with COVID-19 vaccines was observed recently. Pagenkopf and Sudmeyer reported a case of longitudinally extensive transverse myelitis following vaccination with ChadOx1nCoV-19. CSF analysis presented predominantly polymorphonuclear pleocytosis [50]. No pathogen was detected in CSF. There were no positive findings for anti-neuronal autoantibodies. A positive SARS-CoV-2-IgG serum antibody was detected without an IgG-antibody response against nucleocapsid antigen. Symptoms improved after high dose corticoid therapy. The case report proved a close association between the COVID-19 vaccine and transverse myelitis. Inflammatory response and autoimmunity due to cytokine storm have been proposed as a possible cause of vaccine related myelitis [49, 51].
4.2.2. Post-vaccinal Meningitis and Encephalitis
A healthy 22-year-old man presented with a first seizure 6 days after the second dose of mRNA-1273 [9]. The patient received a lumbar puncture for CSF analysis. SARS-CoV-2 spike S1 receptor-binding domain (RBD) IgG was detected in CSF at a concentration of 98.54 binding antibody units (BAU)/ml and in serum at a concentration of 3270.22 BAU/ml. SARS-CoV-2 spike S1 domain antibody in CSF suggests a possible association with postvaccine complications [9]. There have been some case reports of autoimmune encephalitis involving ChadOx1nCoV-19 [52, 53] and BNT162b2 [54, 55]. Zuhorn et al. reported a case series of 3 patients whose onset of encephalitis was within 7 to 11 days [52]. The temporal association is in line with our case. Baldelli et al. reported a case of hyperacute encephalopathy following ChAdOx1 nCoV-19 vaccine administration [53]. Cytokine levels documented a significant increase in interleukin (IL)-6 in both CSF and se-rum and increased IL-8 in CSF. Saito et al. reported a case of a 42-year-old female who had severe headache and fever after vaccination with BNT162b2. Sp SARS-CoV2 IgG was detected (17.1 AU/mL) in CSF [55]. Elevated serum Sp SARS-CoV-2 IgG (393.4 AU/mL; cut-off 0–50 AU/mL; (Chemiluminescent Enzyme Immunoassay; Architect Quant IgG II, Abbott) was noted. These cases responded well to systemic steroids. Vojdania and Kharrazian conducted research that measured the degree of immune reactivity of anti-SARS-CoV-2 spike protein monoclonal antibody and anti-SARS-CoV-2 nucleoprotein monoclonal an-tibody with human antigens [56]. Vojdania et al. further applied human monoclonal an-ti-SARS-CoV-2 antibodies (spike protein, nucleoprotein) and rabbit polyclonal an-ti-SARS-CoV-2 antibodies (envelope protein, membrane protein) to 55 different tissue antigens [57]. Human anti-SARS-CoV-2 spike protein antibody reacted strongest with neurofilament protein (NFP), followed by mitochondrial M2 antigen, glutamic acid decarboxylase 65 (GAD-65), and nuclear antigen [57]. This study found immune reactivity between SARS-CoV-2 antibodies and barrier target proteins (occludin+zonulin, beta-catenin, and S100B), which were responsible for the integrity of the barriers. Hussain et al. generated an adult mouse model and proved that suppressing endothelial Wnt/β-catenin signalling causes increased blood‒brain barrier permeability [58]. This demonstrated that β-catenin-dependent Wnt signalling maintains the integrity of the adult blood‒brain bar-rier. CTNNB1 encodes β-catenin. Tran et al. used an inducible and conditional knockout (icKO) mouse model with tamoxifen-inducible endothelial cell-restricted disruption of CTNNB1, and the iCKO mice developed severe seizures, neuronal injury, and central nervous system inflammation [59]. The immune reactivity between SARS-CoV-2 antibodies and brain barrier protein is responsible for the neurological complications of vaccination. There are some mechanisms of COVID-19 encephalitis. Autoimmune encephalitis and systemic inflammatory status responses were two same mechanisms between vaccine and SARS-COV-2 related encephalitis [60, 61]. Direct virus invasion was the different mechanism of SARS-COV-2 induced encephalitis from post-vaccinal encephalitis. SARS-COV-2 PCR positivity has been detected in the CSF in a small number of patients [62]. The resulting incidence of encephalitis following ChadOx1nCoV-19 was 0.08 per 100,000 PY and 0.02 following BNT162b2 [63]. It is essential to recognize and provide ap-propriate treatment for this complication.
4.2.3. Vaccine-associated herpes simplex encephalitis
Several case reports have revealed the effects of SARS-CoV-2 vaccinations on herpes simplex virus and varicella-zoster virus reactivation [64-72]. Cases of herpetic meningoencephalitis following COVID-19 vaccines of ChAdOx1 nCoV-19 [73] and Covaxin [74] were reported. However, the actual incidence of vaccine-associated herpes simplex encephalitis was not well established. Transient lymphopenia was observed in participants vaccinated with BNT162b1 [75]. Toll-like receptors, plasmacytoid dendritic cells, interferon, natural killer cells, macrophages [76], CD4 and CD8 T cells [77] contribute to the im-mune response against HSV [78]. Lymphocyte depletion may be related to primary herpes or reactivation of latent herpes [79, 80]. The immunomodulation process has been report-ed as a possible cause of herpes virus reactivation after vaccination [81]. Higher frequency of herpes reactivation was observed during COVID-19. This may be explained by immune dysregulation including CD4 cell and NK cell cytopenia, stimulation of IL-6 and tumour necrosis factor-α [82]. Further studies on the mechanism of susceptibility to herpesvirus infection after COVID-19 vaccination can help clarify the association.
4.2.4. Stroke
The COVID-19 vaccine causes the platelet-activating antibodies, against Platelet fac-tor 4 (PF4). The onset of thrombosis at unusual sites such as in the brain or abdomen and thrombocytopenia occurring after 5 to 20 days after vaccination can present a rare adverse effect of ChAdOx1 nCov‐19 vaccination [83]. The detailed molecular pathway was described by Goldman M, Hermans C [15]. Positive charge PF4 bind to negative surface of the adenoviral vector hexons and formed an immunogenic complex and triggered anti-PF4 antibodies. Macrophages, monocytes, Natural killer cells, and dendritic cells will take up antibody-PF4 (adenovirus) complexes via FC gamma receptors, resulting in the activation of the cells. Platelets bind to such complex via their Fc gamma RIIA and activated release of PF4 which resulted in thrombosis. Another mechanism was related to soluble spike protein. Soluble spike protein bind to ACE2 receptor and may trigger immunological events and induce thromboembolic events [84]. This mechanism has termed “Vaccine-Induced COVID-19 Mimicry” syndrome and may explain the correlation of adenovirus-based vaccine and thromboembolic events [84]. SARS-CoV-2 and adenovirus bind and activate various Toll-like receptors (TLRs), leading to activation of the NFκB signalling pathway and further released phosphorylated p50/p65 heterocomplex to the nucleus where transcription of genes including TNFα, IL-1, IL-6, MCP-1, MCP-3, ICAM-1, VCAM-1, complement components and coagulation factors, such as PAI-1 is induced [85-87]. ACE2 activity is lost during SARS-CoV-2 viral entry [88]. The loss of ACE2 activity allowed Ang II mediated activation of p38 MAPK activation and simultaneously promoting thrombosis [89]. P38 MAPK pathway regulates the translation of TNF-α and IL-1β which also activated NF-κB pathway [90]. NF-κB pathway involved in inflammatory and thrombotic response [91]. Ischemic stroke, haemorrhagic stroke and Cerebral venous sinus thrombosis (CVST) were reported after COVID-19 vaccines [92]. Ischemic strokes are associated with VITT. Haemorrhagic stroke was usually secondary to CVST [93]. One case report revealed a case without cardiovascular risk factors, and with no remarkable past medical history, suffered from large primary haemorrhagic stroke after ChAdOx1 nCoV‐19 vaccination without thrombocytopenia, thrombosis, or aneurysm [94]. VITT was not the only mechanism of vaccine related intracranial haemorrhage (ICH). CVST is a rare form of stroke and often occurred in young and middle-aged woman [95]. According to an analysis of cases reported to the European Medicines Agency about CVST [96], more cases were reported after ChAdOx1 nCov‐19 vaccination than after mRNA vaccination. Antibodies against PF4 were reported in 8% of cases in the ChAdOx1 nCov‐19 group but in none in the mRNA vaccination group. Thrombocytopenia was reported in 57% of CVST cases in the ChAdOx1 nCov‐19 group. CVST occurring after ChA-dOx1 nCov‐19 has different associated laboratory data from the mRNA vaccination group [96]. Thrombocytopenia was only reported in the ChAdOx1 nCov‐19 group and pre‐COVID‐19 CVST group. No case was reported in the mRNA vaccine group. Kowarz et al. reported molecular evidence that vector-based vaccines were associated with soluble spike protein variants [84]. Production of soluble spike protein was caused by splicing events. Soluble spike protein has been found to cause an inflammatory response in endothelial cells [14, 97]. A population-based cohort study estimated the risk of CVST after Ad26.COV2.S [98]. The overall age- and sex-adjusted CVST incidence post–Ad26.COV2.S vaccination was 2.34 per 100,000 PY. Age-adjusted CVST rates for female and male individuals were 2.46 per 100,000 PY and 2.34 per 100,000 PY, respectively. Another web‐based questionnaire focused on the incidence of cerebral sinus and venous thrombosis within one month after COVID‐19 vaccination in 2021 in 9 German states [99]. The incidence rate was 1.52 (95% CI = 1.00–2.21) per 100,000 person‐months for ChAdOx1, 0.11 per 100,000 person‐months for BNT162b2, and 0.00 (95% CI = 0.00–1.48) per 100,000 per-son‐months for mRNA‐1273. Although CVST is rare, prompt identification and initiated treatment is crucial. CVST with VITT appeared with higher rates of coma, intracranial haemorrhage and high mortality rate [100]. Current treatment for VITT include high dose immune globulin, corticosteroid, rituximab, eculizumab and therapeutic plasma exchange. Non-heparin anticoagulants, including fondaparinux, direct oral anticoagulants (DOACs), and direct thrombin inhibitors (argatroban, bivalirudin) were suggested [101].
Point 4: Conclusion is that neurological complications are rare in general. However, there are no frequencies mentioned either in your scarcely described cohort or in general literature. Since SARS-related diseases are endothelial diseases, with an increase in stroke and myocardial infarction possibly related to activation of ACE receptors, it is also necessary to discuss pathophysiological aspects: how likely is it, that both COVID-19 and spike protein-related vaccination complications produce vascular and immunological complications via the same mechanism?
Thank you for the suggestion. We have amended the manuscript in the conclusion:
Anti- severe acute respiratory syndrome coronavirus 2 (SARS-CoV-2) spike protein IgG detected in CSF and elevated serum Spike-specific (Sp) SARS-CoV-2 IgG concentration may indicated that the central nervous system adverse events were induced after vaccination [11, 12]. SARS-CoV-2 spike protein which expressed after either mRNA or adenoviral-vector vaccines, binds to the angiotensin-converting enzyme 2 (ACE2) receptor followed by endocytosis. Neuropathologic findings may be due to direct endothelial cell damage cause by SARS-COV-2 spike protein which was due to molecular mimicry elicited by the spike protein [13]. The autoimmune response with complement activation and in-creased cytokine expression may cause disruption of blood-brain barrier and the concomitant thromboses/hypercoagulable state [14]. Thrombosis combined with thrombocytopenia after vaccination has been termed vaccine-induced prothrombotic immune thrombocytopenia (VIPIT), VITT or thrombosis with thrombocytopenia syndrome (TTS). VITT has been reported after immunization with adenovirus vector-based vaccines against COVID-19 mostly [15-19]. The mean age of patients with VITT was 45.6 years (95% CI 43.8-47.4, I2=57%), with a female predominance (70%) [17]. Another systemic review presented with the median age of 45.5 years [19]. One systemic review which included 27 patients with occurrence or relapse of Thrombotic thrombocytopenic purpura (TTP) [18]. The mean age was 51.3 years. TTP was mostly seen after BNT162b2 vaccine (after first dose, n = 12; after second dose, n = 7), followed by mRNA-1273 vaccine (after first dose, n = 2; after second dose, n = 4). Adenoviral vaccines were responsible for 4 cases (Ad26.COV2-S - 1, ChAdOx1 nCoV-19 - 3). These were the most common mechanisms of neurological complications of COVID-19 vaccines. Age seems to be a poor prognostic factor of neurological adverse events. A systemic review noted that older age, higher modified Rankin scale (MRS) and second dose of vaccine presented a worse outcome of patient with COVID-19 vaccine myelitis [20]. According to Sisonke study of of Ad26.COV2.S , the median age of participants were 42 years and the majority were women. Adverse events decreased with increasing age (3.2% for age 18–30 years, 2.1% for age 31–45 years, 1.8% for age 46–55 years, and 1.5% for age > 55 years) [21]. Decline of innate immune defence mechanisms in older people may be the reason. Older patients tend to have lower systemic levels of IL-6, IL-10, C-reactive protein, and lower neutralising antibody titres after vaccination as compared to younger individuals [22].
Last, we are deeply honored by the time and effort you spent in reviewing this manuscript. In reviewing and revising our manuscript, we are motivated to read more and thus learn more from your criticisms.

Reviewer 2 Report
Major comment:
1. The authors had published a “similar report”, on Vaccines, which seems more comprehensive than the current one, although it was focused on Acute Encephalitis. To be more compact and acceptable, it should contain a Literature Review even for those within the same countries, as you mentioned in lines 40-45, the data is already recorded!
2. Abstract and conclusion should be expanded to convey the manuscript's interesting content
3. As well, the keywords MUST be expanded to contain the parameters you already mentioned in the manuscript title, at least.
4. Similar to the first comment, totally the mechanisms behind these neurological complications of post-covid vaccination you mentioned, seem very primitive and limited. There are numerous reports analyzing this issue plus neurological complications accompanied by SARS-CoV-2 infection too and suggest multiple mechanisms, please find (at least in PubMed), compare, and cite them.
5. Lines 61-70, material and methods, do you recommend citing the references used for diagnosis of all diseases you diagnosed? Including Herpes encephalitis.
6. Out of your 19 cases, 10 cases are >=60, is this considered age as a risk factor?!, please discuss this issue in line with the international publications.
7. Do you have any discussion for Bell’s palsy that appeared in 7 out of 19 cases, why it appears after the first, second, third, and in one case after the fourth dose?
Author Response
Response to Reviewer 2 Comments
Point 1: The authors had published a “similar report”, on Vaccines, which seems more comprehensive than the current one, although it was focused on Acute Encephalitis. To be more compact and acceptable, it should contain a Literature Review even for those within the same countries, as you mentioned in lines 40-45, the data is already recorded!
Author Reply
We are deeply honored by the time and effort you spent in reviewing this manuscript. We have revised the manuscript thoroughly according to your suggestions. The title was also changed to “Investigation of Neurological Complications after COVID-19 Vaccination: Report of the Clinical Scenarios and Review of the Literature”. We added more literature review sections for the rarity and frequency of these neurological complications. In the section of discussion, all changes have been marked on yellow background. The responses to your comments are below:
The incidence rate of Bell’s palsy was about 6.07 per 100,000 persons (95% CI, 3.49–8.65) after the first dose of mRNA vaccination and decreased to 5.06 per 100,000 persons (95% CI, 1.97–8.16) after the second dose of mRNA vaccination.
The incidence rate of Guillain‒Barré Syndrome (GBS) after mRNA vaccination was 0.19 per 100,000 persons (95% CI, 0.13–0.25) [31]. Narasimhalu et al. reported a rare case of acute trigeminal neuritis and cervical radiculitis after Pfizer-BioNtech vaccination [42].
The actual incidence of trigeminal neuralgia was not well known. Demyelinating disease of the central nervous system include multiple sclerosis (MS), myelin oligodendrocyte glycoprotein (MOG) antibody associated disease (MOGAD), neuromyelitis optica spectrum disorder (NMOSD), and acute disseminated encephalomyelitis (ADEM) [44, 45]. These disorders can cause TM [46]. NMOSD was reported after mRNA vaccines (86.56%), inactivated vaccines (11.94%), and viral vector vaccines (1.49%) [47]. MOGAD was less commonly. A 59-year-old male had MOGAD with longitudinal extensive transverse myelitis after ChadOx1nCoV-19 [48]. A multinational network cohort study estimated incidence rates of transverse myelitis. The incidence was ranging from 1-4 per 100,000 PY in meta-analysis, depend on age and sex strata [32]. Transverse myelitis associated with COVID-19 vaccines was observed recently. Pagenkopf and Sudmeyer reported a case of longitudinally extensive transverse myelitis following vaccination with ChadOx1nCoV-19. CSF analysis presented predominantly polymor-phonuclear pleocytosis [50]. No pathogen was detected in CSF. There were no positive findings for anti-neuronal autoantibodies. A positive SARS-CoV-2-IgG serum antibody was detected without an IgG-antibody response against nucleocapsid antigen. Symptoms improved after high dose corticoid therapy. The case report proved a close association between the COVID-19 vaccine and transverse myelitis. Inflammatory response and autoimmunity due to cytokine storm have been proposed as a possible cause of vaccine related myelitis [49, 51]. A population-based cohort study estimated the risk of CVST after Ad26.COV2.S [98]. The overall age- and sex-adjusted CVST incidence post–Ad26.COV2.S vaccination was 2.34 per 100,000 PY. Age-adjusted CVST rates for female and male individuals were 2.46 per 100,000 PY and 2.34 per 100,000 PY, respectively. Another web‐based questionnaire focused on the incidence of cerebral sinus and venous thrombosis within one month after COVID‐19 vaccination in 2021 in 9 German states [99]. The incidence rate was 1.52 (95% CI = 1.00–2.21) per 100,000 person‐months for ChAdOx1, 0.11 per 100,000 person‐months for BNT162b2, and 0.00 (95% CI = 0.00–1.48) per 100,000 person‐months for mRNA‐1273.
Point 2: Abstract and conclusion should be expanded to convey the manuscript's interesting content.
Author Reply
Thank you for the suggestion. We have amended the manuscript in abstract and conclusion as follows:
Abstract: Coronavirus disease 2019 (COVID-19), caused by severe acute respiratory syndrome coronavirus 2 (SARS-CoV-2), broke out in 2019 and became a pandemic in 2020. Since then, vaccines have been approved to prevent severe illness. However, vaccines are associated with the risk of neurological complications ranging from mild to severe. Severe complications like vac-cine-induced immune thrombotic thrombocytopenia (VITT) associated acute ischemic stroke have been reported as a rare complication post COVID-19 vaccination. During pandemic era, VITT evaluation is needed in case of history of vaccination within last one month prior to the event. Cerebral venous sinus thrombosis (CVST) should be suspected in patients following immunization with persistent headache and unresponsive to analgesics. In this article, we investigated the neurological complications after COVID-19 vaccination and provide more subsequent related clinical studies of accurate diagnosis, pathophysiological mechanisms, incidence, outcome, and management.
Conclusions: Although most neurological complications are nonfatal, there are some fatal complications, such as CVST, GBS, ADEM and ICH. Both COVID-19 and spike protein-related vaccines produce immunological complications through the same mechanism of molecular mimicry. VITT is another unique mechanism. Clinicians should be aware of these possible complications and made differential diagnosis thoroughly. Earl Recognition and appropriate treatments should be initiated once these adverse events occur. Future investigations will be required to identify the risk factor of neurological complications of different COVID vaccines and technologically advanced new vaccines against the unfavourable adverse effects.
Point 3: As well, the keywords MUST be expanded to contain the parameters you already mentioned in the manuscript title, at least.
Author Reply
Thank you for the suggestion. We have amended the keywords:
COVID-19 vaccines; neurological complications; Bell’s palsy, Guillain‒Barré syndrome, transverse myelitis, post-vaccinal meningitis and encephalitis, vaccine-associated herpes simplex encepha-litis, cerebral venous sinus thrombosis.
Point 4: Similar to the first comment, totally the mechanisms behind these neurological complications of post-covid vaccination you mentioned, seem very primitive and limited. There are numerous reports analyzing this issue plus neurological complications accompanied by SARS-CoV-2 infection too and suggest multiple mechanisms, please find (at least in PubMed), compare, and cite them.
Author Reply
Thank you for the suggestion and comments. We have amended the manuscript in the discussion:
Anti- severe acute respiratory syndrome coronavirus 2 (SARS-CoV-2) spike protein IgG detected in CSF and elevated serum Spike-specific (Sp) SARS-CoV-2 IgG concentration may indicated that the central nervous system adverse events were induced after vaccina-tion [11, 12]. SARS-CoV-2 spike protein which expressed after either mRNA or adenovi-ral-vector vaccines, binds to the angiotensin-converting enzyme 2 (ACE2) receptor fol-lowed by endocytosis. Neuropathologic findings may be due to direct endothelial cell damage cause by SARS-COV-2 spike protein which was due to molecular mimicry elicited by the spike protein [13]. The autoimmune response with complement activation and in-creased cytokine expression may cause disruption of blood-brain barrier and the concom-itant thromboses/hypercoagulable state [14]. Thrombosis combined with thrombocytopenia after vaccination has been termed vaccine-induced prothrombotic immune thrombocytopenia (VIPIT), VITT or thrombosis with thrombocytopenia syndrome (TTS). VITT has been reported after immunization with adenovirus vector-based vaccines against COVID-19 mostly [15-19]. The mean age of patients with VITT was 45.6 years (95% CI 43.8-47.4, I2=57%), with a female predominance (70%) [17]. Another systemic review presented with the median age of 45.5 years [19]. One systemic review which included 27 patients with occurrence or relapse of Thrombotic thrombocytopenic purpura (TTP) [18]. The mean age was 51.3 years. TTP was mostly seen after BNT162b2 vaccine (after first dose, n = 12; after second dose, n = 7), followed by mRNA-1273 vaccine (after first dose, n = 2; after second dose, n = 4). Adenoviral vaccines were responsible for 4 cases (Ad26.COV2-S - 1, ChAdOx1 nCoV-19 - 3).
4.1. Peripheral Nervous System Disorder
4.1.1 Bell’s Palsy
Inflammation and oedema of the facial nerve is the mechanism of facial nerve palsy. Inflammation-caused demyelination [23], anatomical variations of fallopian canal, cold or viral prodrome, vasospasm related primary ischaemia, transudate associated secondary ischaemia, thickening of facial nerve sheath [24], HSV infection [25] and Interferon [26] are possible aetiologies of Bell’s palsy. The actual mechanism of Bell’s palsy after vaccination is unclear. Cytokine storm and direct invasion were the possible mechanism which made SARS-COV-2 the possible aetiology of Bell’s palsy [27]. There were some FDA's Vaccines and Related Biologic Products Advisory Committee, and a sponsor (Pfizer) raised the possibility of innate immune activation from a combined effect of mRNA and lipids [28]. One case report disclosed a possible association between BNT162b2 and autoimmune reaction. IgG autoantibodies against type I IFNs increased compared to pre-vaccination, second and third shots [29]. A higher frequency of facial paralysis after the Pfizer-BioNTech vaccine than after the Oxford-AstraZeneca vaccine was observed based on EudraVigilance data [30]. According to a large-scale meta-analysis which disclosed a marginal increase of incidence of Bell’s palsy after mRNA vaccinations compared to the unvaccinated groups, the incidence rate of Bell’s palsy was about 6.07 per 100,000 persons (95% CI, 3.49–8.65) after the first dose of mRNA vaccination and decreased to 5.06 per 100,000 persons (95% CI, 1.97–8.16) after the second dose of mRNA vaccination. The risk of Bell’s palsy after BNT162b2 was significantly higher than after mRNA-1273 [31]. Another cohort study reported that the incidence of Bell’s palsy was 15-101 per 100,000 person years in meta-analysis, depend on age and sex strata [32]. In our case series report, most patients with Bell’s palsy received mRNA-1273.
4.1.2 GBS
There are some case reports of GBS being associated with the COVID-19 vaccines BNT162b2 [33] and ChadOx1nCoV-19 [34]. According to an interim analysis of surveillance of COVID-19 vaccines (Ad.26.COV2. S, BNT162b2, or mRNA-1273 COVID-19 vaccine), elevated risk of GBS after primary Ad.26.COV2. S was confirmed, and mRNA vaccines do not appear to be associated with GBS [35]. According to a meta-analysis, the incidence rate of GBS after mRNA vaccination was 0.19 per 100,000 persons (95% CI, 0.13–0.25) [31]. The risk of GBS of BNT162b2 was significantly higher than that of mRNA-1273 (POR, 2.85; 95% CI, 1.61–5.04) with low heterogeneity (I2, 0%). One multinational network cohort study estimated the incidence of GBS which ranged from 1-12 per 100,000 per-son-years (PY) in meta-analysis stratified by age and sex [32]. The casual mechanism of GBS post COVID-19 vaccines was not established. SARS-COV-2 entered host cells via S protein, which not only binds to ACE2 receptor but also binds to sialic acid–containing glycoproteins and gangliosides on cell surfaces [36]. Post-infectious immune-mediated mechanism was hypothesized because anti-ganglioside antibodies were recognized in several COVID-19 patients [37, 38]. It may be hypothesized that the SARS-CoV-2 spike-bearing gangliosides may form a neoantigen recognized by unidentified antibodies [38]. Cytokine storm was another possible mechanism of GBS induced by COVID [39]. Current treatment for GBS include immune globulin and plasma exchange [40]. Some patients died; another didn’t substantially improve [40]. In our case series, there were no Guillain‒Barré syndrome cases associated with COVID-19 vaccines. This may be due to limited by sample size.
4.1.3 Trigeminal Neuralgia
Trigeminal neuralgia is a rare neurological complication of COVID-19 vaccines. Immune-mediated inflammation is thought to be a more likely mechanism due to its rapid onset and response to corticosteroids and pregabalin [41-43]. Narasimhalu et al. reported a rare case of acute trigeminal neuritis and cervical radiculitis after Pfizer-BioNtech vaccination [42]. The 52-year-old female had a past medical history of diabetes mellitus, hypertension, hyperlipidaemia, and scoliosis. Her symptoms started 3 hours after the first dose of Pfizer-BioNtech vaccination and improved gradually after oral prednisolone. To date, the actual incidence of trigeminal neuralgia is not well known.
4.2. Central Nervous System
4.2.1 Demyelinating disease of the central nervous system
Transverse myelitis (TM) is inflammation of the spinal cord and causes acute or subacute neurologic signs of motor, sensory or autonomic dysfunction. Demyelinating disease of the central nervous system include multiple sclerosis (MS), myelin oligodendrocyte glycoprotein (MOG) antibody associated disease (MOGAD), neuromyelitis optica spectrum disorder (NMOSD), and acute disseminated encephalomyelitis (ADEM) [44, 45]. These disorders can cause TM [46]. NMOSD was reported after mRNA vaccines (86.56%), inactivated vaccines (11.94%), and viral vector vaccines (1.49%) [47]. MOGAD was less commonly. A 59-year-old male had MOGAD with longitudinal extensive transverse myelitis after ChadOx1nCoV-19 [48]. In case14, the patient received high-dose steroids for 5 days, and her condition improved. This patient recovered partially. Case 13 responded poorly to steroids, which may be due to late initiation of the steroid treatment. One multi-analysis included 37 cases of transverse myelitis following different vaccines including those against hepatitis B virus, measles-mumps, rubella vaccine, diphtheria–tetanus–pertussis (DTP), diphtheria–tetanus (DT), rabies vaccine, oral polio virus, influenza vaccine, typhoid vaccine, pertussis, and Japanese B Encephalitis [49]. A multinational network cohort study estimated incidence rates of transverse myelitis. The incidence was ranging from 1-4 per 100,000 PY in me-ta-analysis, depend on age and sex strata [32]. Transverse myelitis associated with COVID-19 vaccines was observed recently. Pagenkopf and Sudmeyer reported a case of longitudinally extensive transverse myelitis following vaccination with ChadOx1nCoV-19. CSF analysis presented predominantly polymorphonuclear pleocytosis [50]. No pathogen was detected in CSF. There were no positive findings for anti-neuronal autoantibodies. A positive SARS-CoV-2-IgG serum antibody was detected without an IgG-antibody response against nucleocapsid antigen. Symptoms improved after high dose corticoid therapy. The case report proved a close association between the COVID-19 vaccine and transverse myelitis. Inflammatory response and autoimmunity due to cytokine storm have been proposed as a possible cause of vaccine related myelitis [49, 51].
4.2.2. Post-vaccinal Meningitis and Encephalitis
A healthy 22-year-old man presented with a first seizure 6 days after the second dose of mRNA-1273 [9]. The patient received a lumbar puncture for CSF analysis. SARS-CoV-2 spike S1 receptor-binding domain (RBD) IgG was detected in CSF at a concentration of 98.54 binding antibody units (BAU)/ml and in serum at a concentration of 3270.22 BAU/ml. SARS-CoV-2 spike S1 domain antibody in CSF suggests a possible association with postvaccine complications [9]. There have been some case reports of autoimmune encephalitis involving ChadOx1nCoV-19 [52, 53] and BNT162b2 [54, 55]. Zuhorn et al. reported a case series of 3 patients whose onset of encephalitis was within 7 to 11 days [52]. The temporal association is in line with our case. Baldelli et al. reported a case of hyperacute encephalopathy following ChAdOx1 nCoV-19 vaccine administration [53]. Cytokine levels documented a significant increase in interleukin (IL)-6 in both CSF and se-rum and increased IL-8 in CSF. Saito et al. reported a case of a 42-year-old female who had severe headache and fever after vaccination with BNT162b2. Sp SARS-CoV2 IgG was detected (17.1 AU/mL) in CSF [55]. Elevated serum Sp SARS-CoV-2 IgG (393.4 AU/mL; cut-off 0–50 AU/mL; (Chemiluminescent Enzyme Immunoassay; Architect Quant IgG II, Abbott) was noted. These cases responded well to systemic steroids. Vojdania and Kharrazian conducted research that measured the degree of immune reactivity of anti-SARS-CoV-2 spike protein monoclonal antibody and anti-SARS-CoV-2 nucleoprotein monoclonal an-tibody with human antigens [56]. Vojdania et al. further applied human monoclonal an-ti-SARS-CoV-2 antibodies (spike protein, nucleoprotein) and rabbit polyclonal an-ti-SARS-CoV-2 antibodies (envelope protein, membrane protein) to 55 different tissue antigens [57]. Human anti-SARS-CoV-2 spike protein antibody reacted strongest with neurofilament protein (NFP), followed by mitochondrial M2 antigen, glutamic acid decarboxylase 65 (GAD-65), and nuclear antigen [57]. This study found immune reactivity between SARS-CoV-2 antibodies and barrier target proteins (occludin+zonulin, beta-catenin, and S100B), which were responsible for the integrity of the barriers. Hussain et al. generated an adult mouse model and proved that suppressing endothelial Wnt/β-catenin signalling causes increased blood‒brain barrier permeability [58]. This demonstrated that β-catenin-dependent Wnt signalling maintains the integrity of the adult blood‒brain bar-rier. CTNNB1 encodes β-catenin. Tran et al. used an inducible and conditional knockout (icKO) mouse model with tamoxifen-inducible endothelial cell-restricted disruption of CTNNB1, and the iCKO mice developed severe seizures, neuronal injury, and central nervous system inflammation [59]. The immune reactivity between SARS-CoV-2 antibodies and brain barrier protein is responsible for the neurological complications of vaccination. There are some mechanisms of COVID-19 encephalitis. Autoimmune encephalitis and systemic inflammatory status responses were two same mechanisms between vaccine and SARS-COV-2 related encephalitis [60, 61]. Direct virus invasion was the different mechanism of SARS-COV-2 induced encephalitis from post-vaccinal encephalitis. SARS-COV-2 PCR positivity has been detected in the CSF in a small number of patients [62]. The resulting incidence of encephalitis following ChadOx1nCoV-19 was 0.08 per 100,000 PY and 0.02 following BNT162b2 [63]. It is essential to recognize and provide ap-propriate treatment for this complication.
4.2.3. Vaccine-associated herpes simplex encephalitis
Several case reports have revealed the effects of SARS-CoV-2 vaccinations on herpes simplex virus and varicella-zoster virus reactivation [64-72]. Cases of herpetic meningoencephalitis following COVID-19 vaccines of ChAdOx1 nCoV-19 [73] and Covaxin [74] were reported. However, the actual incidence of vaccine-associated herpes simplex encephalitis was not well established. Transient lymphopenia was observed in participants vaccinated with BNT162b1 [75]. Toll-like receptors, plasmacytoid dendritic cells, interferon, natural killer cells, macrophages [76], CD4 and CD8 T cells [77] contribute to the im-mune response against HSV [78]. Lymphocyte depletion may be related to primary herpes or reactivation of latent herpes [79, 80]. The immunomodulation process has been report-ed as a possible cause of herpes virus reactivation after vaccination [81]. Higher frequency of herpes reactivation was observed during COVID-19. This may be explained by immune dysregulation including CD4 cell and NK cell cytopenia, stimulation of IL-6 and tumour necrosis factor-α [82]. Further studies on the mechanism of susceptibility to herpesvirus infection after COVID-19 vaccination can help clarify the association.
4.2.4. Stroke
The COVID-19 vaccine causes the platelet-activating antibodies, against Platelet fac-tor 4 (PF4). The onset of thrombosis at unusual sites such as in the brain or abdomen and thrombocytopenia occurring after 5 to 20 days after vaccination can present a rare adverse effect of ChAdOx1 nCov‐19 vaccination [83]. The detailed molecular pathway was described by Goldman M, Hermans C [15]. Positive charge PF4 bind to negative surface of the adenoviral vector hexons and formed an immunogenic complex and triggered anti-PF4 antibodies. Macrophages, monocytes, Natural killer cells, and dendritic cells will take up antibody-PF4 (adenovirus) complexes via FC gamma receptors, resulting in the activation of the cells. Platelets bind to such complex via their Fc gamma RIIA and activated release of PF4 which resulted in thrombosis. Another mechanism was related to soluble spike protein. Soluble spike protein bind to ACE2 receptor and may trigger immunological events and induce thromboembolic events [84]. This mechanism has termed “Vaccine-Induced COVID-19 Mimicry” syndrome and may explain the correlation of adenovirus-based vaccine and thromboembolic events [84]. SARS-CoV-2 and adenovirus bind and activate various Toll-like receptors (TLRs), leading to activation of the NFκB signalling pathway and further released phosphorylated p50/p65 heterocomplex to the nucleus where transcription of genes including TNFα, IL-1, IL-6, MCP-1, MCP-3, ICAM-1, VCAM-1, complement components and coagulation factors, such as PAI-1 is induced [85-87]. ACE2 activity is lost during SARS-CoV-2 viral entry [88]. The loss of ACE2 activity allowed Ang II mediated activation of p38 MAPK activation and simultaneously promoting thrombosis [89]. P38 MAPK pathway regulates the translation of TNF-α and IL-1β which also activated NF-κB pathway [90]. NF-κB pathway involved in inflammatory and thrombotic response [91]. Ischemic stroke, haemorrhagic stroke and Cerebral venous sinus thrombosis (CVST) were reported after COVID-19 vaccines [92]. Ischemic strokes are associated with VITT. Haemorrhagic stroke was usually secondary to CVST [93]. One case report revealed a case without cardiovascular risk factors, and with no remarkable past medical history, suffered from large primary haemorrhagic stroke after ChAdOx1 nCoV‐19 vaccination without thrombocytopenia, thrombosis, or aneurysm [94]. VITT was not the only mechanism of vaccine related intracranial haemorrhage (ICH). CVST is a rare form of stroke and often occurred in young and middle-aged woman [95]. According to an analysis of cases reported to the European Medicines Agency about CVST [96], more cases were reported after ChAdOx1 nCov‐19 vaccination than after mRNA vaccination. Antibodies against PF4 were reported in 8% of cases in the ChAdOx1 nCov‐19 group but in none in the mRNA vaccination group. Thrombocytopenia was reported in 57% of CVST cases in the ChAdOx1 nCov‐19 group. CVST occurring after ChA-dOx1 nCov‐19 has different associated laboratory data from the mRNA vaccination group [96]. Thrombocytopenia was only reported in the ChAdOx1 nCov‐19 group and pre‐COVID‐19 CVST group. No case was reported in the mRNA vaccine group. Kowarz et al. reported molecular evidence that vector-based vaccines were associated with soluble spike protein variants [84]. Production of soluble spike protein was caused by splicing events. Soluble spike protein has been found to cause an inflammatory response in endothelial cells [14, 97]. A population-based cohort study estimated the risk of CVST after Ad26.COV2.S [98]. The overall age- and sex-adjusted CVST incidence post–Ad26.COV2.S vaccination was 2.34 per 100,000 PY. Age-adjusted CVST rates for female and male individuals were 2.46 per 100,000 PY and 2.34 per 100,000 PY, respectively. Another web‐based questionnaire focused on the incidence of cerebral sinus and venous thrombosis within one month after COVID‐19 vaccination in 2021 in 9 German states [99]. The incidence rate was 1.52 (95% CI = 1.00–2.21) per 100,000 person‐months for ChAdOx1, 0.11 per 100,000 person‐months for BNT162b2, and 0.00 (95% CI = 0.00–1.48) per 100,000 per-son‐months for mRNA‐1273. Although CVST is rare, prompt identification and initiated treatment is crucial. CVST with VITT appeared with higher rates of coma, intracranial haemorrhage and high mortality rate [100]. Current treatment for VITT include high dose immune globulin, corticosteroid, rituximab, eculizumab and therapeutic plasma exchange. Non-heparin anticoagulants, including fondaparinux, direct oral anticoagulants (DOACs), and direct thrombin inhibitors (argatroban, bivalirudin) were suggested [101].
Point 5: Lines 61-70, material and methods, do you recommend citing the references used for diagnosis of all diseases you diagnosed? Including Herpes encephalitis.
Author Reply
Facial motor impairment should be characterized as Bell palsy only after the exclusion of traumatic, iatrogenic, neoplastic, infectious, metabolic, and congenital aetiologies. Electroneuronography (ENoG) and electromyography (EMG) are the primary tests used for prognosticating facial nerve outcomes [3]. Peripheral neuropathy was diagnosed by symptoms and nerve conduction study (NCS) findings. Motor and sensory conduction velocity can be measured of peripheral nerves. In axonal neuropathies, conduction velocity is normal or reduced mildly and needle electromyography of denervation in affected muscles. In demyelinating neuropathies, nerve conduction velocities may be slowed con-siderably in affected fibers, or conduction is blocked completely [4]. The diagnosis of transverse myelitis was made based on “Proposed diagnostic criteria and nosology of acute transverse myelitis” [5]. The diagnosis was confirmed by T2 hyperintense signal change on spinal magnetic resonance imaging and exclusion of a compressive cord lesion. Lumbar puncture was administered for differential diagnosis. The diagnosis of encephalitis was made based on symptoms such as altered mental status, seizure, fever, focal neurological findings, and cerebrospinal fluid (CSF) analysis or neuroimaging [6]. Herpes encephalitis was diagnosed based on herpes simplex virus (HSV) polymerase chain reaction (PCR) analysis of CSF is the gold standard method for the diagnosis of HSV encephalitis [7]. CVST was confirmed by magnetic resonance venography (MRV) [8]
Point 6: Out of your 19 cases, 10 cases are >=60, is this considered age as a risk factor?! please discuss this issue in line with the international publications.
Author Reply
Thank you for the suggestion and comments. We have amended the manuscript in discussion as follows:
Age seems to be a poor prognostic factor of neurological adverse events. A systemic review noted that older age, higher modified Rankin scale (MRS) and second dose of vaccine presented a worse outcome of patient with COVID-19 vaccine myelitis [20]. According to Sisonke study of of Ad26.COV2.S, the median age of participants were 42 years and the majority were women. Adverse events decreased with increasing age (3.2% for age 18–30 years, 2.1% for age 31–45 years, 1.8% for age 46–55 years, and 1.5% for age > 55 years) [21]. Decline of innate immune defence mechanisms in older people may be the reason. Older patients tend to have lower systemic levels of IL-6, IL-10, C-reactive protein, and lower neutralising antibody titres after vaccination as compared to younger individuals [22].
Point 7: Do you have any discussion for Bell’s palsy that appeared in 7 out of 19 cases, why it appears after the first, second, third, and in one case after the fourth dose?
Author Reply
Thank you for the suggestion and comments. We have amended the manuscript as follows:
In this case studies, most Bell’s palsy presented after the first or second doses of COVID-19 vaccination, which is also consistent with the report of Lai, Y et al. [31]. It is not common for adverse events appeared after the fourth dose of vaccine. Only one of our cases had Bell’s palsies after the fourth dose of vaccination.
Reference
- Lai, Y. H., H. Y. Chen, H. H. Chiu, Y. N. Kang, and S. B. Wong. "Peripheral Nervous System Adverse
Events after the Administration of Mrna Vaccines: A Systematic Review and Meta-Analysis of Large-Scale
Studies." Vaccines (Basel) 10, no. 12 (2022).
Last, we are deeply honored by the time and effort you spent in reviewing this manuscript. In reviewing and revising our manuscript, we are motivated to read more and thus learn more from your criticisms.

Round 2
Reviewer 1 Report
Thank you for your thoughtful and extensive review.
Reviewer 2 Report
thank you very much, now it seems more acceptable.